

# Projections of East Asian summer monsoon change at global warming of 1.5°C and 2°C

Jiawei Liu[1,2], Haiming Xu[1,2] and Jiechun Deng[1,2]

[1]Collaborative Innovation Center on Forecast and Evaluation of Meteorological Disasters /KLME / ILCEC, Nanjing University of Information Science and Technology, Nanjing, China

[2]College of Atmospheric Sciences, Nanjing University of Information Science and Technology, Nanjing, China

*Correspondence to*: H. Xu (hxu@nuist.edu.cn)

**Abstract.** Much research is needed regarding two long-term warming targets of the 2015 Paris Agreement, i.e., 1.5°C and 2°C above pre-industrial levels, especially from a regional perspective. The East Asian summer monsoon (EASM) intensity and associated precipitation changes under both warming targets are explored in this study. Multimodel ensemble mean projections by 19 CMIP5 models show small increases in EASM intensity and general increases in summer precipitation at 1.5° and 2°C warming, but with large multimodel standard deviations. Thus, a novel multimodel ensemble pattern regression (EPR) method is applied to give more reliable projections based on the concept of "emergent constraints," which is effective to tighten the range of multimodel diversity and harmonize the changes of different variables over the EASM region. Future changes projected by using the EPR method suggest decreased precipitation over the Meiyu belt and increased precipitation over the high latitudes of East Asia and central China, together with a considerable weakening of EASM intensity. Furthermore, suppressed precipitation would appear over 30°-40°N of East Asia in June and over the Meiyu belt in July, with enhanced precipitation at their north and south sides. These changes in early summer are attributed to a southeastward retreat of western North Pacific high (WNPSH) and a southward shift of East Asian subtropical jet (EASJ), which weaken the moisture transport via southerly wind at low level and alter vertical motions over the EASM region. In August, precipitation would increase over the high latitudes of East Asia with more moisture from the wetter area over the ocean in the east and decrease over Japan with westward extension of WNPSH. These monthly precipitation changes would finally contribute to a tripolar pattern of EASM precipitation change at 1.5°and 2°C warming. Corrected EASM intensity exhibits a slight difference between 1.5°C and 2°C, but a pronounced moisture increase during extra 0.5°C leads to enhanced EASM precipitation over large areas in East Asia at 2°C warming.

## 1. Introduction

The East Asian summer monsoon (EASM) is one of the most important climate systems, which provides a plenty of summer rainfall to densely populated areas in East Asia. Variation of the EASM exerts great influences on flooding, drought and agricultural harvest, and thus has close relationships with livelihoods of billions of people in countries including China, Japan and South Korea (Huang et al., 2012). As global





warming is ongoing, much attention has been paid to the EASM change in a warmer climate (e.g., Chen and Sun, 2013; Seo et al., 2013).

To limit anthropogenic influences on climate systems, the 2015 Paris Agreement under the United Nations Framework Convention on Climate Change (UNFCCC) includes two long-term global temperature goals, i.e., "holding the increase in the global average temperature to well below 2°C above pre-industrial levels and pursuing efforts to limit the temperature increase to 1.5°C" (UNFCCC, 2015). While a world of 2°C warming is inadequate to be considered safe (Structured Expert Dialogue (SED), 2015), there is a lack of scientific research regarding the 1.5°C target, especially for regional climates (Mitchell et al., 2016). Studies generally evaluated differences of global climate impacts at global warming of 1.5°C and 2°C levels (Schleussner et al., 2016; Sanderson et al., 2017). No clear pictures, however, are presented for the EASM under both scenarios.

Many studies projected EASM intensity and precipitation for different time periods in the 21st century (e.g., Chen and Sun, 2013; Seo et al., 2013). For the near future (2016–2035), projections suggest EASM precipitation amount and intensity would have statistically significant increases over most regions of East Asia (Chen and Sun, 2013). By the end of the 21st century, a much more obvious increase in precipitation is projected, although some pattern differences exist under different Representative Concentration Pathway scenarios (RCPs; Chen and Sun, 2013; Seo et al., 2013). In spite of a general increase in summer precipitation over East Asia in these projections, summer monsoon would only slightly strengthen due to increased zonal and meridional land-sea thermal contrasts over the 21st century (Jiang and Tian, 2013). For specific global warming targets, most attention was paid to the projections of climate extremes, including heat waves and precipitation extremes over China (Guo et al., 2016, 2017).

Although most projections of EASM change are based on coupled ocean–atmosphere general circulation models (CGCMs) from the Intergovernmental Panel on Climate Change (IPCC) fifth phase of the Coupled Model Intercomparison Project (CMIP5) (e.g., Chen and Sun, 2013; Seo et al., 2013), significant intermodel spreads exist in these simulations (Huang et al., 2013; Chen and Bordoni, 2014).



Similarly, projections of some climate systems, which are highly related to the EASM variability, have severe uncertainty. The East Asian subtropical jet (EASJ) stream, which induces warm advection and leads to upward moisture transport, has a close relationship with the EASM rainfall (Sampe and Xie, 2010; Kosaka et al., 2011). Thus, future position of rain belt may become much more uncertain due to the

meridional position biases of EASJ in CGCMs (Ma et al. 2015). On the other hand, the western North Pacific subtropical high (WNPSH) also shows a great diversity in CMIP5 simulations (He and Zhou, 2015). The WNPSH transports moisture via southerly wind on its western flank and anchor the rain belt on its northwestern periphery (Zhou and Yu, 2005).

The multimodel ensemble mean (MMM) method and model weighting method are commonly applied to improve the reliability of climate model projections. However, these methods have problems with common background bias among ensemble members (e.g., Li and Xie, 2014; Wang et al. 2014; Huang and Ying, 2015; Li et al., 2016), leading to biases in MMM change and limiting the reliability of future projections (e.g., Boé et al. 2009; Cox et al. 2013). Another method based on the concept of "emergent

constraints" is used in future climate projections to reduce uncertainties (e.g., Boé et al., 2009; Räisänen et al., 2010; Abe et al., 2011; Bracegirdle and Stephenson, 2012; Cox et al., 2013; Huang and Ying, 2015; Li et al., 2016). This method detects relationship between intermodel similarity in the observed climate and that in the simulated future climate change. Calibrations by this "present-future" relationship can efficiently constrain the simulated climate change to a narrower range (Boé et al. 2009; Bracegirdle and

Stephenson 2012; Huang and Ying 2015; Li et al., 2016).

In this study, we investigate the EASM change at levels of 1.5°C and 2°C global mean surface air temperature (GMT) increases above pre-industrial levels (henceforth 1.5°C and 2°C warming), using state-of-the-art CGCM outputs from the IPCC CMIP5. The influences of biases related to EASJ and

WNPSH on EASM precipitation change are examined first. Then, projections of EASM intensity and precipitation at 1.5°C and 2°C warming are given using a novel "emergent constraints" correction method. The rest of the paper is organized as follows. In Sect. 2, we describe datasets and methods. The





intermodel spread is examined in Sect. 3. We present major results of corrected EASM changes in Sect. 4.

A summary and discussion are given in Sect. 5.

## 2. Data and Methods

### 2.1 Data

In this study, 43 available CMIP5 CGCMs are first examined in terms of their performances of EASM

precipitation simulation for the reference period of 1986-2005. With the skill score S (Taylor, 2001) of

precipitation over the EASM region (100°-150°E, 20°-50°N) greater than 0.75, 19 CMIP5 CGCMs are

selected for further analyses, including their conventional variables in historical and RCP4.5 (a medium

emission scenario) runs (Taylor et al., 2012). A brief summary of the models is given in Table 1. For both

historical and RCP4.5 runs, monthly data from first ensemble member (r1i1p1) are used in our analyses.

Precipitation from the Global Precipitation Climatology Project (GPCP; Adler et al. 2003), sea surface

temperature (SST) from Hadley Centre Sea Ice and Sea Surface Temperature (HadISST1.1; Rayner et al.,

2003) dataset and several variables from the ECMWF ERA-Interim reanalysis (Dee et al., 2011) for the

reference period are also used in this study. All datasets from observations and models are interpolated to

1°×1° grid first.

### 2.2 Methods

Periods of 1.5°C and 2°C GMT increase above pre-industrial levels are defined as 20-year time slices

relative to the reference period (Table 1). As the reference period is 0.61°C warmer than pre-industrial

levels of 1850–1900 (IPCC, 2013), 1.5°C and 2°C warmings translate to 0.89 and 1.39°C above the

reference period levels, respectively, following Schleussner et al. (2016).

Several indexes have been designed to measure the strength of the EASM. According to the assessment

of Wang et al. (2008), the shear vorticity index defined by Wang and Fan (1999) is best correlated with

the leading principle component (PC) of EASM multivariate Empirical Orthogonal Function (EOF)

analysis on a set of six meteorological fields (the correlation coefficient is -0.97). Therefore, to measure

the EASM intensity change, we adopt negative of the Wang and Fan (1999) index (WFN)





$$WFN = (U_{850}; 110°E - 140°E, 22.5°N - 32.5°N) - (U_{850}; 90°E - 130°E, 5°N - 15°N). \qquad (1)$$

Following the recommendation of He et al. (2015), 500-hPa eddy geopotential height ($H_e$) is used to measure the WNPSH in warming climate rather than traditionally used geopotential height. $H_e$ is defined

as the deviation of geopotential height from the regional average over 0°–40°N globally and the $H_e=0$ contour represents the boundary of WNPSH. Because of the increase in 500-hPa geopotential height over the entire East Asia and western North Pacific, the East Asian summer rain belt does not follow the traditional indicator, the 5880-m contour of 500-hPa geopotential height over the western North Pacific (He et al., 2015). A more suitable indicator, $H_e$, was thsu designed (He et al., 2015) to solve this problem.

The multimodel ensemble pattern regression (EPR) method proposed by Huang and Ying (2015) is used to correct biases of future precipitation change in individual models and MMM. A brief introduction to this method is provided here. Future change ($C_i$) is defined as the difference of future climatology ($F_i$) and historical climatology ($H_i$) in model i, i.e., $C_i = F_i - H_i$. The future change ($C_i$) can be decomposed into the real change ($C_{real}$), common change bias ($\bar{C}' = N^{-1}\sum_{i=1}^{N}C_i - C_{real}$) and individual change bias

($C_i'' = C_i - N^{-1}\sum_{i=1}^{N}C_i$), respectively, as follows,

$$C_i = C_{real} + \bar{C}' + C_i'' \qquad (2)$$

Similarly, the historical climatology ($H_i$) can be decomposed into the observed climate ($H_{obs}$), common historical bias ($\bar{H}' = N^{-1}\sum_{i=1}^{N}H_i - H_{obs}$) and individual historical bias ($H_i'' = H_i - N^{-1}\sum_{i=1}^{N}H_i$),

respectively, as below,

$$H_i = H_{obs} + \bar{H}' + H_i'' \qquad (3)$$

Then, spatially correlated modes between the historical and future change biases are explored by using the intermodel diversity in $H''$ and $C''$ from all the models. To be specific, the EOF analysis is

performed on $H''$, and spatially orthogonal modes $EOF_j$ with corresponding $PC_{ij}$, where j=1, …, M are obtained. Here, M equals 12 or 16 for different variables, which is adequate to represent $H''$ and $\bar{H}'$. The present-future relationship is established through a multivariant linear regression analysis on PCs and $C''$. Thus, $C''$ can be estimated as follows:





$$\hat{C}_i'' = \sum_j^M \hat{b}_j PC_{ij}, \tag{4}$$

where $\hat{b}_j$ denotes regression patterns.

$\bar{H}'$ can be further projected onto $EOF_j$:

$$\bar{H}' = \sum_j^M \text{EOF}_j e_j, \tag{5}$$

where $e_j$ denotes expansion coefficients.

Substituting $e_j$ into Eq. (4), an estimation of $\bar{C}'$ can be expressed by:

$$\hat{\bar{C}}' = \sum_j^M \hat{b}_j e_j. \tag{6}$$

Thus, the corrected MMM change can be estimated as $\bar{C}_c = \bar{C} - \hat{\bar{C}}'$, where $\bar{C} = N^{-1} \sum_{i=1}^N C_i$. Similarly, individual model changes can also be corrected.

### 3. Intermodel Spread

Although the periods of 1.5°C and 2°C warming projected are not far away from present (Table 1), a

15 general increase in summer precipitation would appear over the EASM region for both warming targets (Figs. 1a, b), consistent with previous findings. Prominent increase is mainly located over the high latitudes (~40°N) covering Northeast China and Korea and the low latitudes (~25°N) covering Southeast China and the western North Pacific. The spatial pattern of precipitation change at 2°C warming is similar to that at 1.5°C, with a larger enhancement for the 2°C warming (Fig. 1). Considering that the

20 precipitation changes range from -2.69 to 4.53 mm/day and from -1.31 to 2.00 mm/day per degree of GMT increase for 1.5°C and 2°C warming, the mutimodel standard deviation ranges from 0.4 to larger than 1.2 mm/day per degree of GMT increase over major precipitation increased areas (Figs. 1a, b), indicative of a large uncertainty in EASM precipitation projection. Figure 2 displays changes in EASM intensity by the 19 individual CMIP5 models and their MMM, measured by the WFN index. Changes in

25 the WFN index for individual models vary from below -2 to above 2, representing significant weakening and strengthening of EASM intensity. Thus, the MMM EASM intensity shows a slight change, which is consistent with Jiang and Tian (2013).



The intermodel variability of summer-mean 200-hPa zonal wind climatology is examined at 1.5°C warming over the EASM region by performing an intermodel EOF analysis on the 19 CMIP5 CGCMs. The first intermodel EOF mode of 200-hPa zonal wind, explaining 59.8% of the total intermodel variability, shows opposite signs to the north and south of ~37°N over East Asia, which captures the uncertainty in meridional position of EASJ (Fig. 3a). Positive and negative values of the first intermodel PC (PC1) indicate northward and southward shifts of EASJ meridional position, respectively (Fig. 3b). Meridional displacement of the EASJ manifests as one of the most dominant modes of upper-tropospheric zonal and meridional wind anomalies along the Asian subtropical jet in summer (Hong and Lu, 2016). This uncertainty may inherit from meridional position biases of EASJ in the historical runs (Ma et al. 2015).

To further investigate the influence of EASJ meridional position uncertainty on the summer precipitation change at 1.5°C warming, we use composite anomalies of precipitation change due to the northward (Fig. 4a; six models; PC1 in Fig. 3b larger than 100) and southward-shifted (Fig. 4b; six models; PC1 in Fig. 3b smaller than -70) EASJ relative to their MMM, and their differences (Fig. 4c). When the EASJ is northward shifted (Fig. 4a), a positive anomaly of precipitation change occurs over the mid latitudes (30°-40°N) of China and a negative anomaly of precipitation change occurs over 20°-25°N of the western North Pacific. On the other hand, a negative anomaly is over 20°-40°N in China due to the southward shift of EASJ (Fig. 4b). The opposite precipitation changes in EASJ northward and southward shifted conditions lead to a statistically significant difference over 20°-40°N in China (Fig. 4c). Similar results are found under 2°C warming (not shown). These results indicate that the meridional shift of EASJ exerts a great influence on the projected EASM precipitation change at both warming targets.

Figure 5 shows a similar intermodel EOF analysis on 500-hPa $H_e$ climatology for the 15 CMIP5 CGCMs at 1.5°C warming. The first intermodel EOF mode (Fig. 5a), explaining 56.7% of the total $H_e$ variability, suggests that the WNPSH would also exhibit major uncertainty in its meridional position, shown as opposite variations to the north and south of ~20°N over the western North Pacific. Positive and





negative values of PC1 (Fig. 5b) represent the southward and northward shifts of the WNPSH boundary, respectively. Figure 6 depicts composite anomalies of precipitation change under southward (Fig. 6a; five models; PC1 in Fig. 5b larger than 300) and northward (Fig. 6b; five models; PC1 in Fig. 5b smaller than -300) shifted WNPSH conditions, respectively, relative to their MMM; and their difference is given

in Fig. 6c. When the WNPSH is southward shifted (Fig. 6a), the southwest-northeast oriented positive precipitation anomaly lies over the western North Pacific and the negative anomaly appears over 35°-45°N in China. The opposite anomalies can be found under the northward shift of WNPSH condition (Fig. 6b), leading to significant differences from those under the southward shifted condition (Fig. 6c). Similar results are found at 2°C warming (not shown). In the literatures, much attention has been paid to

the change of WNPSH under ongoing global warming, but projections based on model outputs are inconclusive (Liu et al., 2014; He and Zhou, 2015; He et al., 2015). Thus, the intermodel uncertainty shown here confirms the difficulty in projecting the WNPSH.

In general, considerable uncertainty exists in projections of EASM intensity, precipitation and major

climate systems at 1.5°C and 2°C warming. To get more reliable projections, a more effective correction method is needed beyond the traditional methods.

**4. Projections Improved by Using the EPR Method**

Figure 7 illustrates the precipitation change improved by the EPR method (Huang and Ying, 2015). The multimodel standard deviation of precipitation decreases dramatically below 0.30 and 0.15 mm/day per

degree of GMT increase over most of the EASM region at 1.5°C and 2°C warming, respectively (Figs. 7a, b), indicative of improved similarity and reliability in projections. Major improvements of precipitation change are located over the Meiyu belt covering East China, Korea and Japan. After the corrections, increased precipitation over the Meiyu belt shows an obvious negative change at 1.5°C warming (Fig. 7a), and a stronger reduction is revealed at 2°C warming (Fig. 7b). Meanwhile, the increase in

precipitation is larger in the high latitudes of East Asia and central China (Figs. 7a, b). While the uniform increase pattern of precipitation before corrections is similar to that in the MMM for the near future, the corrected change pattern is in accordance with the result of the six best-performing models in the study of



Chen and Sun (2013). These changes suggest that some wet areas (e.g., the Meiyu belt) may face more droughts and some arid and semi-arid areas (e.g., the high latitudes of East Asia and central China) may get wetter, which may bring challenges in various aspects.

Uncorrected projections show a slight change in MMM EASM intensity at both 1.5°C and 2°C warming, which is consistent with previous studies (e.g., Jiang and Tian, 2013). Figure 8 shows monthly changes in MMM WFN index using the corrected data. The EASM intensity displays a considerable weakening at both 1.5°C and 2°C warming in June, July and August. This change could lead to a weaker moisture transport by the southerly wind over East Asia (Fig. 10) and thus impact the spatial pattern of

precipitation change (Fig. 7).

Considering remarkable differences between the East Asia monsoon in early summer and in late summer (Wang et al., 2010), monthly changes in precipitation and 850-hPa wind from June to August are further illustrated in Fig. 9 to show the seasonal march of summer precipitation. In June (Fig. 9a), precipitation is

reduced by 20-50% over 30°-40°N extending from east China to Korea and Japan. Besides, largely increased precipitation is over the western North Pacific with a southward retreat of the WNPSH boundary. In July (Fig. 9b), precipitation decreases over the Meiyu belt accompanied with an anticyclone wind change, which is more obvious at 2°C warming (Fig. 9e). Meanwhile, enhanced precipitation is located on both north and south sides of the Meiyu belt with the eastward shifted WNPSH boundary.

Thus, a tripolar pattern of precipitation change is formed over East Asia, especially at 2°C warming. The monsoon in southern East Asia is closely related to the WNPSH in early summer (Chen et al., 2004). Southward and eastward shifts of the WNPSH boundary in June and July suggest that more moisture may linger over the western North Pacific, rather than being transported to the mid latitudes of East Asia, in agreement of the weakened EASM intensity (Fig. 8). In August (Fig. 9c), precipitation increases over the

high latitudes (40°-50°N) of East Asia and decreases over Japan, together with a westward extension of the WNPSH. The displacements of the WNPSH in July and August are offset, leading to no significant change in its summer mean position (Figs. 7a, b).





Changes in specific humidity and moisture flux at 850 hPa are presented in Fig. 10. Specific humidity keeps enhancing over almost the whole EASM region under continuous global warming, and an obvious increase occurs at 2°C warming, due to the extra 0.5°C warming. Budget analysis in Seo et al. (2013) revealed that the domain-averaged precipitation increase over East Asia has a tight link with enhanced

evaporation due to increased surface temperature. From this perspective, enhanced specific humidity at 850 hPa fundamentally supports the increase in domain-averaged precipitation over the EASM region. In addition, the spatial patterns of low-level specific humidity enhancement and moisture flux change are consistent with that of the precipitation change (Fig. 10). In June (Fig. 10a), slightly enhanced specific humidity and a southward moisture transport appear over 30°-40°N of East Asia where precipitation is

reduced. On the other hand, a relatively strong increase is located over the western North Pacific and the high latitudes of East Asia where precipitation is increased, accompanied with a relatively strong northward moisture transport. In July (Fig. 10b), the change in specific humidity is quite similar to that in precipitation. The specific humidity increase is relatively small with a weak moisture transport over the Meiyu belt and is much larger with a strong transport of moisture to both north and south sides of the rain

belt. In August (Fig. 10c), specific humidity is slightly increased over the western North Pacific due to the westward extension of the WNPSH. Over the high latitudes of East Asia, specific humidity is robustly enhanced with a westward moisture transport from wet areas over the ocean.

Figure 11 shows meridional sections of monthly wind change averaged over 115°-120°E at 1.5°C and

2°C warming. A prominent feature of EASJ change is the southward shift in early summer. The corresponding vertical motion change is consistent with the precipitation change. In June (Fig. 11a), a descending motion change appears over 30°-40°N and an ascending change motion appears to the south and north. In July (Fig. 11b), a descending motion change is significant at the former vertically-tilted ascending area, which is anchored by EASJ axis during the reference period. Many studies have

addressed the importance of EASJ to the EASM precipitation (e.g., Liao et al., 2004; Sampe and Xie 2010; Kosaka et al. 2011). Therefore, the southward shift of EASJ and the corresponding vertical motion change further determine the tripolar pattern of precipitation change over the EASM region. In August (Fig. 11c), the EASJ strengthens to the north and south of the EASJ core during the reference period. The





vertical transport change induced by the change in EASJ is relatively small. Still, an ascending change is favorable for increased precipitation over the high latitudes of East Asia.

## 5. Summary and Discussion

In this study, we present the changes in EASM intensity and the associated precipitation projected by 19
CMIP5 CGCMs, and examine the influences of two most related climate systems' biases on the EASM precipitation change pattern at global warming levels of 1.5°C and 2°C. Using the "emergent constraints" method, we provide more reliable projections of EASM intensity and precipitation.

Although MMM projections for 1.5°and 2°C warming exhibit a general increase in summer precipitation and a slight change in EASM intensity, which is in accordance with previous findings (e.g., Chen and Sun, 2013; Jiang and Tian, 2013; Seo et al., 2013), of the most importance is that large model uncertainty cannot be ignored in these projections. A large multimodel standard deviation appears along with change in precipitation, and the projected EASM intensity experiences huge diversity among individual models. In addition, uncertainties in projected meridional positions of both EASJ and WNPSH, captured by an intermodel EOF analysis, have significant influences on the change in East Asia summer precipitation.

Given the limitation of the traditional methods in climate change projections, a novel EPR method (Huang and Ying, 2015) based on the concept of "emergent constraints" is used to provide more reliable projections. The multimodel standard deviation of precipitation is largely reduced over the entire EASM region after the correction. Prominent corrections include decreased precipitation over the Meiyu belt and increased precipitation over the high latitudes of East Asia and central China. Additionally, the EASM intensity is considerably weakened in June, July and August at 1.5°and 2°C warming, which agrees with the decreased northward temperature gradient over East Asia (not shown). Monthly change projections further suggest that reduced precipitation over 30°-40°N of East Asia in June and over the Meiyu belt in July are mainly determined by changes in the EASJ and WNPSH. Southeastward retreatment of the WNPSH and southward shift of EASJ act to weaken the moisture transport via

southerly wind at low level and hinder the vertical ascending motion over the reduced-precipitation area. Moreover, precipitation increases on both north and south sides of reduced-precipitation area, which is beneficial by the wetter environment and the ascending motion change. In early summer, precipitation increases more over southern East Asia, resulting from a weakened northward moisture transport. In

August, a robust increase in precipitation is over the high latitudes of East Asia with sufficient moisture from the wetter area over the ocean in the east, where surface temperature is greatly increased (not shown), and suppressed precipitation is located over Japan with a westward extension of the WNPSH. Precipitation changes in each month of summer finally form a tripolar pattern of EASM precipitation change at 1.5°C and 2°C warming.

Regarding to the differences between 1.5°C and 2°C warming scenarios, corrected EASM intensity displays a slight change. Enhanced summer precipitation over large area of East Asia may be caused by a pronounced moisture increase during 2°C warming thanks to the extra 0.5°C warming. However, the descending motion change induced by the EASJ change overwhelms the effect of moisture increase over

the Meiyu belt.

Climate models are useful tools for climate projections, and their basic performances in simulating EASM precipitation were demonstrated in many literatures (e.g., Zhou and Zou 2010; Zhou et al. 2013). However, significant biases in simulated location, amount and seasonal evolution of precipitation over

East Asia exist (e.g., Zhou et al. 2009; Huang et al., 2013; Chen and Bordoni, 2014). In this study, we aim at projecting more reliable future changes in EASM intensity and precipitation by using the "emergent constraints" strategy. In our results, changes in various variables have much less multimodel diversity, and show improved consistency over the EASM region.

*Competing interests.* The authors declare no competing interests.

*Acknowledgements.* We acknowledge the World Climate Research Programme's Working Group on



Coupled Modeling, which is responsible for the CMIP5, and the climate modelling groups for producing

and making available their model outputs. This work was jointly supported by the National Natural

Science Foundation of China (grants 41490643 and 41575077), and the Priority Academic Program

Development of Jiangsu Higher Education Institutions (PAPD).

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



Table 1. Summary of 19 CMIP5 models used in this study and the time periods for 1.5°C and 2°C

warming above pre-industrial levels.

| Model Name | Institution(s) | 1.5°C | 2°C |
|---|---|---|---|
| ACCESS1-0 | CSIRO-BOM | 2018-2037 | 2040-2059 |
| ACCESS1-3 | CSIRO-BOM | 2020-2039 | 2039-2058 |
| BNU-ESM | BUN | 2013-2032 | 2032-2051 |
| CanESM2 | CCCma | 2014-2033 | 2028-2047 |
| CCSM4 | NCAR NSF-DOE-NCAR | 2024-2043 | 2052-2071 |
| CESM1-CAM5 | NCAR NSF-DOE-NCAR | 2018-2037 | 2034-2053 |
| CMCC-CM | CMCC | 2024-2043 | 2041-2060 |
| CMCC-CMS | CMCC | 2021-2040 | 2039-2058 |
| CNRM-CM5 | CNRM-CERFACS | 2029-2048 | 2050-2069 |
| CSIRO-Mk3-6-0 | CSIRO-QCCCE | 2023-2042 | 2035-2054 |
| GFDL-CM3 | NOAA-GFDL | 2008-2027 | 2022-2041 |
| IPSL-CM5A-LR | IPSL | 2019-2038 | 2035-2054 |
| IPSL-CM5A-MR | IPSL | 2015-2034 | 2035-2054 |
| IPSL-CM5B-LR | IPSL | 2028-2047 | 2053-2072 |
| MPI-ESM-LR | MPI-M | 2024-2043 | 2054-2073 |
| MPI-ESM-MR | MPI-M | 2027-2046 | 2047-2066 |
| MRI-CGCM3 | MPI-M | 2033-2052 | 2064-2083 |
| NorESM1-M | NCC, NMI | 2028-2047 | 2060-2079 |
| NorESM1-ME | NCC, NMI | 2028-2047 | 2053-2072 |



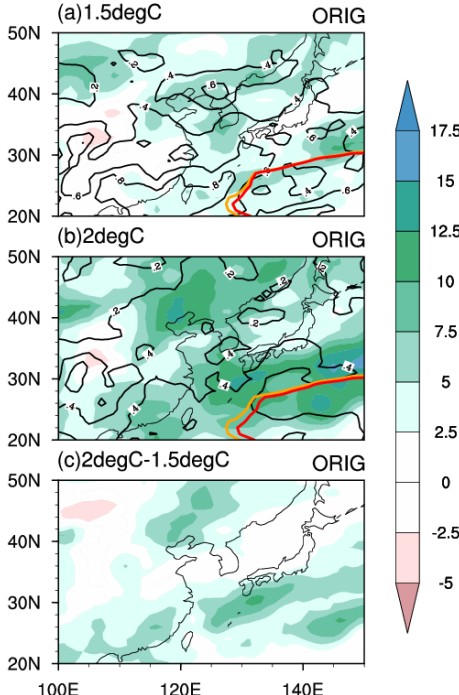

**Figure 1: MMM projected summer precipitation change in percentage (shading; %) and multimodel precipitation change standard deviation (mm/day; black contours) per degree of GMT increase at (a) 1.5°C and (b) 2°C warming relative to the reference period. (c) Difference of (a) and (b). Thick orange curves denote WNPSH boundaries for the reference period, and thick red curves denote MMM-projected WNPSH boundaries at (a) 1.5°C and (b) 2°C warming.**

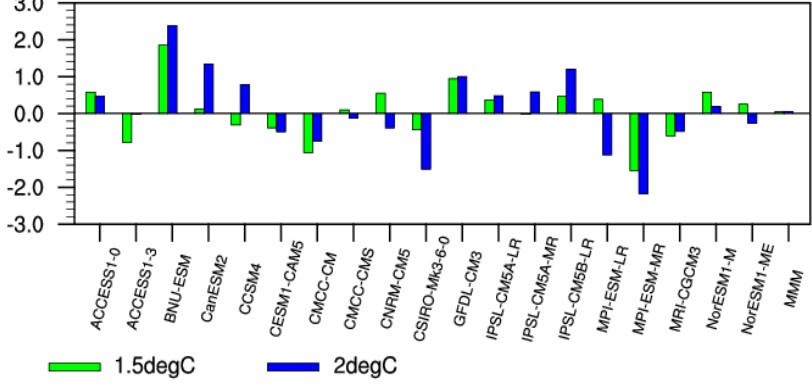



**Figure 2: WFN index change in 19 individual CMIP5 models listed in Table 1, and MMM at 1.5°C (green bars) and 2°C (blue bars) warming relative to the reference period.**

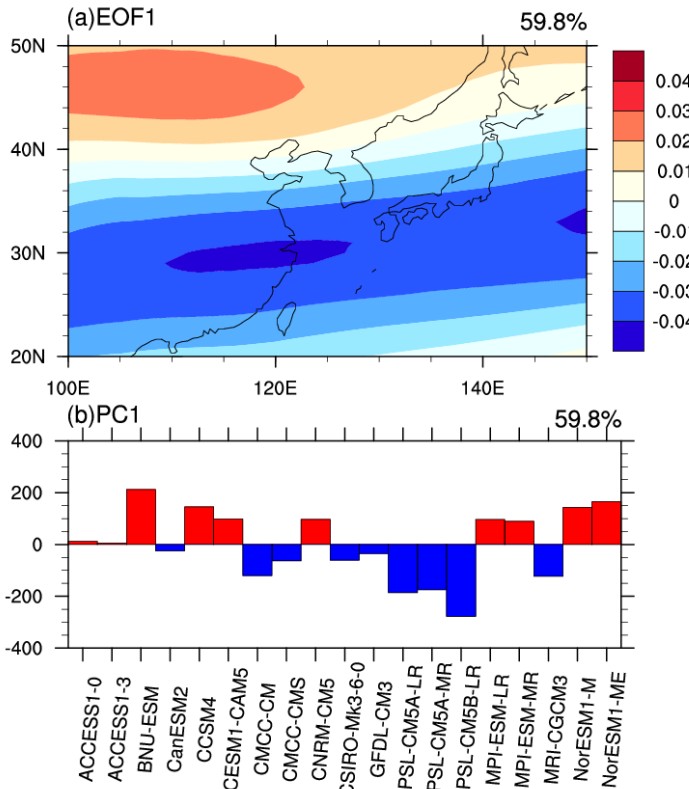

5    **Figure 3: (a) First mode of intermodel EOF analysis (EOF1) of 200-hPa zonal wind (nineteen models) over EASM region (100°-150°E, 20°-50°N) at 1.5°C warming. (b) Corresponding intermodel principal component (PC1) of each model.**





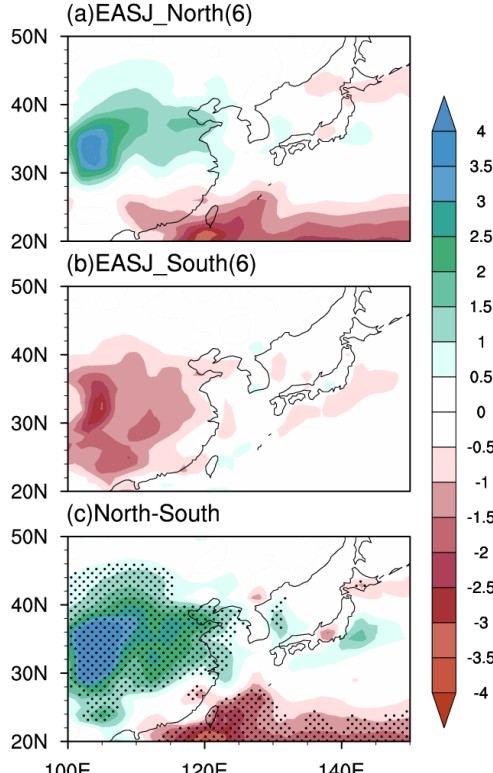

Figure 4: Composites of precipitation anomalies (shading; mm/d) for EASJ (a) northward (six models; PC1 in Fig. 3b larger than 100) and (b) southward (six models; PC1 in Fig. 3b smaller than -70) conditions relative to MMM precipitation change at 1.5°C warming. (c) Difference of (a) and (b). Stippling indicates significance at 95% confidence level.



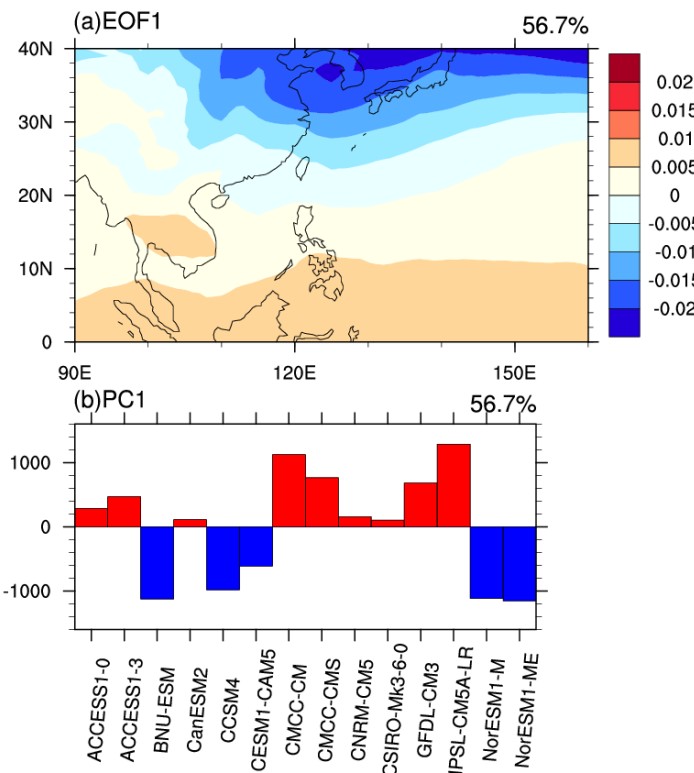

Figure 5: Same as Fig. 3, except for 500-hPa eddy geopotential height ($H_e$; fifteen models).





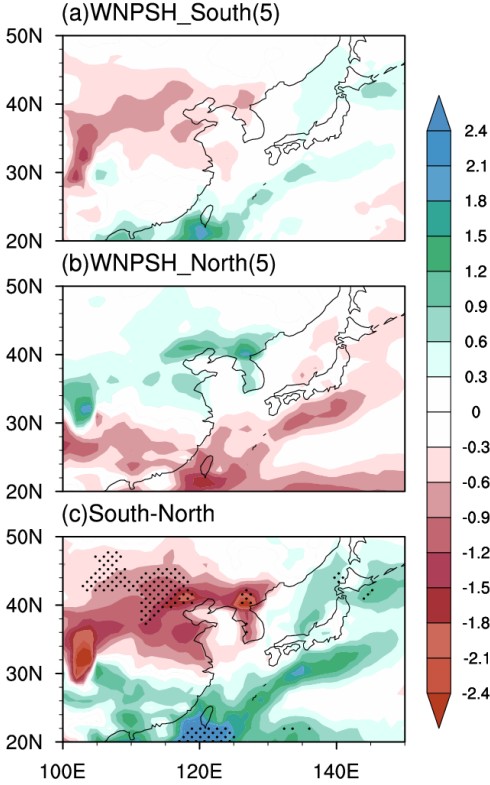

Figure 6: Same as Fig. 4, except for WNPSH (a) southward (five models; PC1 in Fig. 5b larger than 300) and

(b) northward (five models; PC1 in Fig. 5b smaller than -300) conditions.





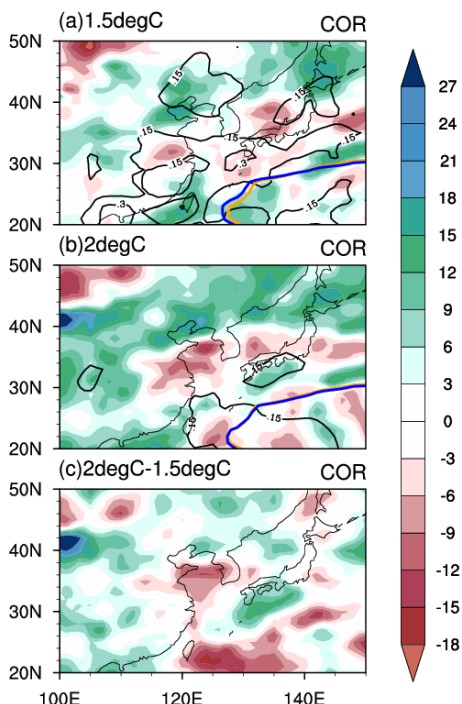

**Figure 7: Same as Fig. 1, except for corrected MMM precipitation change percentage (shading; %). Blue contours denote corrected MMM-projected WNPSH boundaries.**

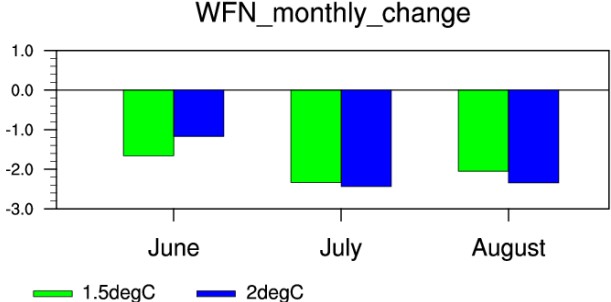

5    **Figure 8: Corrected monthly MMM WFN index change at 1.5°C (green bars) and 2°C (blue bars) warming relative to the reference period.**



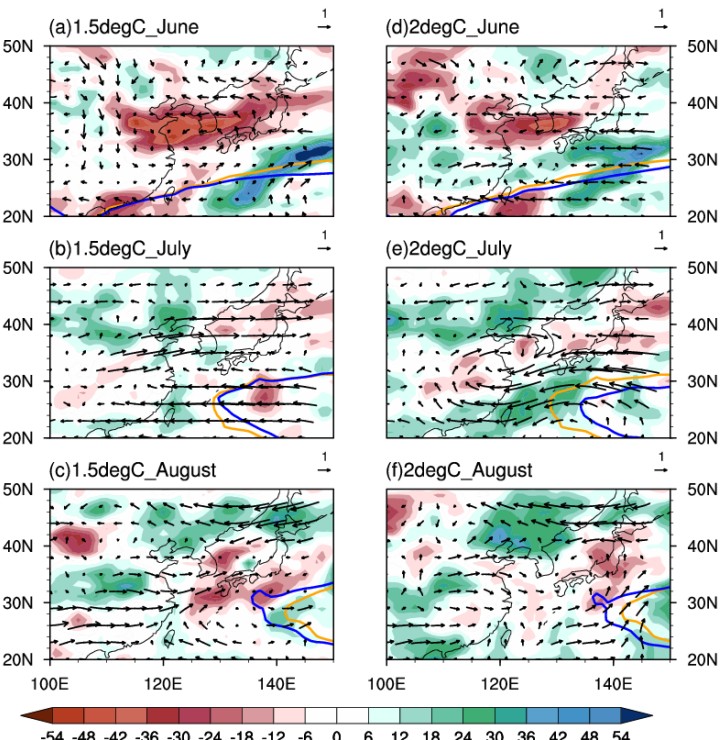

**Figure 9: Corrected monthly precipitation change percent (shading; %) and horizontal 850-hPa wind change (vector; m/s) for (a, d) June, (b, e) July and (c, f) August. Left panels: 1.5°C warming relative to the reference period; right panels: 2°C warming. Orange and blue contours denote the boundaries of WNPSH for the reference period and projected warming levels, respectively.**



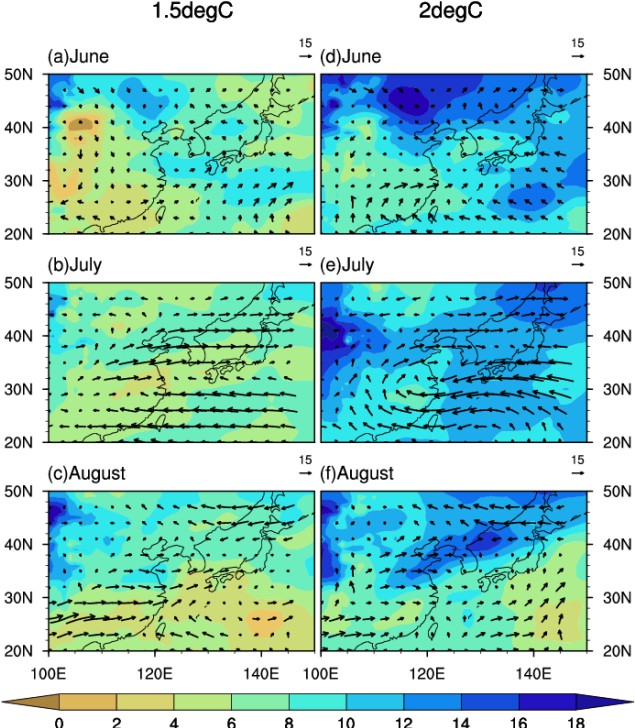

**Figure 10: Corrected monthly 850-hPa specific humidity change percentage (shading; %) and 850-hPa moisture flux change (vector; g kg$^{-1}$ m s$^{-1}$) for (a, d) June, (b, e) July and (c, f) August. Left panels: 1.5°C warming relative to the reference period; right panels: 2°C warming.**



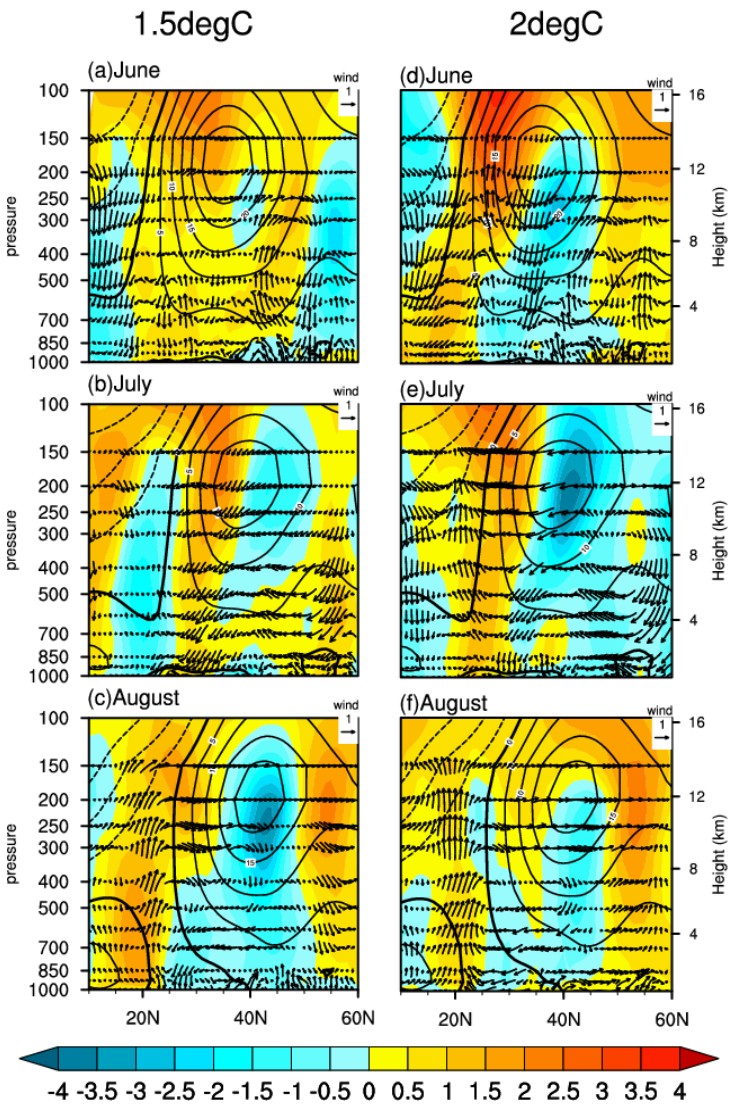

**Figure 11: Meridional sections of corrected monthly zonal wind change (shading; m s-1) and the meridional and vertical wind change (arrow; m s⁻¹ and Pa s⁻¹, respectively; vertical wind is multiplied by 100) averaged from 115°E to 120°E for (a, d) June, (b, e) July and (c, f) August. Left panels: 1.5°C warming relative to the reference period; right panels: 2°C warming. Contours denote climatological-mean zonal wind speed (m s⁻¹) for the reference period.**