# Peer review of "Projections of East Asian summer monsoon change at global warming of 1.5°C and 2°C"

_Earth System Dynamics, 2018_

## Referee Comment (RC1) · Anonymous Referee #1 · 5 Feb 2018

General Comments:

The authors have examined the projected changes in EASM precipitation intensity at two global warming levels of 1.5°C and 2°C using CMIP5 coupled climate models. Further, an ensemble pattern regression (EPR) method is applied to provide more reliable projections. The authors proved that the multimodel standard deviation has reduced drastically after applying the EPR method. The paper then discusses in detail about the monthly variation in projected precipitation changes over East Asia and the related meteorological parameters, during summer monsoon season. The study is interesting as it tries to minimize the inter-model spread in the projections of EASM.

Other comments:

[Figure]

1) Page 3, line 23, and throughout the paper. The usage of GMT can be avoided as it gets confused with Greenwich Mean Time.

2) Page 5. Need some more clarity in explaining the EPR method. Eg. It is difficult to find out what is real change (Creal), how it is computed. The variable N is not defined. Similarly, the variable M, is it the number of models?

3) Page 7, line 25, Is it 14 or 15 models? Why 14 models are used, in Figure 5, instead of 19 models.

---

## Referee Comment (RC2) · Anonymous Referee #2 · 23 Feb 2018

This manuscript analyzes the projections of East Asian summer monsoon (EASM) change at global warming of 1.5°C and 2°C. The authors use a new method named ensemble pattern regression (EPR) to correct the projections of East Asian summer monsoon change. However, there are some severe problems in this manuscript.

Major comment:

1. The authors first emphasize two climate systems (Fig.4 and Fig.6), the East Asian subtropical jet (EASJ) and the western North Pacific subtropical high (WNPSH), which are highly related to the EASM. However, the connection between the changes in these systems are not studied based on the original MME or the corrected changes.

2. The EPR method is used to correct the projection of changes in precipitation, mon-

soon intensity, wind and specific humidity. However, the authors don't show whether the correction of common change bias of each variable is corrected by the historical common bias from itself or not. If so, the connection between the historical bias of one variable and its changes should be demonstrated at first, which is the precondition to use the EPR method.

3. The EPR method introduced by Huang and Ying (2015) depends on some parameters, for example the EOF numbers. These parameters could induce some unreal correction. Thus, it should be carefully chosen and explicitly illustrated to show it is reasonably chosen here.

4. In Section 4, the authors show the intermodel standard deviation (SD) of precipitation decreased dramatically. How about the intermodel SD of changes in specific humidity and wind?

5. This study corrects the projections of EASM change at global warming of 1.5°C and 2°C. The authors conclude the difference of EASM precipitation change between 1.5°C and 2°C is because of the moisture increase by extra 0.5°C. What is meaning to compare the EASM between global warming of 1.5°C and 2°C? The EPR method doesn't seem to help compare their differences.

6. Only 19 models were used in this study. I suggest more models should be used to increase the robustness of multi-model projection.

---

## Author Comment (AC1) · 9 Mar 2018

We appreciate the reviewer's comments and suggestions on our manuscript. Our replies follow each of reviewer's comments or suggestions.

1) Page 3, line 23, and throughout the paper. The usage of GMT can be avoided as it gets confused with Greenwich Mean Time.

**Response**: We have changed "GMT" to "GMST" in our revised manuscript.

2) Page 5. Need some more clarity in explaining the EPR method. Eg. It is difficult to find out what is real change (Creal), how it is computed. The variable N is not defined. Similarly, the variable M, is it the number of models?

**Response**: We agree with the reviewer's suggestion to explain the EPR method more clearly. However, real change (Creal) is just an idealized concept that cannot be computed. What we actually computed was the estimation of common change bias ($\hat{\bar{C}}'$). Then, corrected MMM change is estimated as $\bar{C}_c = \bar{C} - \hat{\bar{C}}'$, where $\bar{C} = N^{-1} \sum_{i=1}^{N} C_i$. Thus, $\bar{C}_c$ can represent $C_{real}$ more reasonably than $\bar{C}$. In addition, $N$ is the number of models and $M$ is the number of EOF modes. We have added some explanations in Section 2.2 Methods as follows:

> *"Although it is impossible to get $C_{real}$, we can close in on it by reducing the bias. And that is what this method tries to do."*
>
> *"Thus, $\bar{C}_c$ can represent $C_{real}$ more reasonably than $\bar{C}$."*

3) Page 7, line 25, Is it 14 or 15 models? Why 14 models are used, in Figure 5, instead of 19 models.

**Response**: We actually used 14 models (fixed) when geopotential height is analyzed. Values of 500-hPa geopotential heights in five models (vary from 2800 to 5300) are much smaller than those in the other models (about 5700 to 5900) in the RPC4.5 runs over the EASM region. Thus, simulations of 500-hPa geopotential height in these five models may not be reliable. Another consideration was that we suspected something was wrong with these data. We obtained the same result, however, after

downloading these data again. As a result, geopotential heights in these five models have been eliminated from our analyses. We have added the following explanations in Section 2.1 Data:

*"Due to intrinsic errors of simulated 500-hPa geopotential height in the RCP4.5 runs of five models, only 14 models are used in geopotential height analysis."*

---

## Author Comment (AC2) · 9 Mar 2018

We appreciate the reviewer's comments and suggestions on our manuscript. Our replies follow each of reviewer's comments or suggestions.

1. The authors first emphasize two climate systems (Fig.4 and Fig.6), the East Asian subtropical jet (EASJ) and the western North Pacific subtropical high (WNPSH), which are highly related to the EASM. However, the connection between the changes in these systems are not studied based on the original MME or the corrected changes.

**Response**: We believe corrected changes of the EASJ and WNPSH have been presented in our previous manuscript, together with their impacts on the EASM change. However, their connections may be not addressed clearly.

As we mentioned in the introduction, the WNPSH acts to transport moisture via southerly wind on its western flank and anchor the rain belt on its northwestern edge (Zhou and Yu, 2005). There is little change in the summer (JJA) mean WNPSH position (Fig. 7), but there are quite obvious change of monthly positions (Fig. 9). Additionally, Fig. 10 shows changes in 850-hPa specific humidity and moisture fluxes. Due to the southward and eastward shifts of the WNPSH boundary in June and July, respectively, more moisture is trapped in the western North Pacific rather than being transported to the mid latitudes of East Asia. In August, the westward extension of the WNPSH boundary leads to larger increase in 850-hPa specific humidity over the high latitudes of East Asia than that over the mid latitudes by strengthening the moisture transport.

The EASJ-induced warm advection leads to upward moisture transport, which has a close relationship with the EASM rainfall (Sampe and Xie, 2010; Kosaka et al., 2011). Thus, change of the EASJ over East Asia is shown in Fig. 11. An obvious southward shift of the EASJ can be found in early summer. Therefore, an anomalous descending motion significantly appears at the former vertically-tilted ascending area anchored by the EASJ axis during the reference period. These changes are consistent with the 850-hPa anomalous anti-cyclone and the decreased precipitation over 30°-40°N of East Asia (Fig. 9).

We have added or revised the related analysis in our revised manuscript as follows:

*"The analyses of corrected WNPSH and moisture transports show consistence between their changes and EASM. Southward and eastward shifts of the WNPSH boundary in early*

*summer lead to a weaker northward moisture transport on its western flank; thus, more moisture and precipitation can be found over the western North Pacific rather than over the mid latitudes of East Asia. In addition, the westward extension of the WNPSH boundary in August leads to a larger increase in 850-hPa specific humidity over the high latitudes of East Asia than that over the mid latitudes, favoring high-latitude precipitation."*

*"Therefore, the southward shift of EASJ and the corresponding vertical motion change in early summer lead to a prominent decrease in precipitation over 30°-40°N of East Asia. These changes correspond to the 850-hPa anomalous anti-cyclone (Fig. 9) and further determine the tripolar pattern of precipitation change over the EASM region."*

2. The EPR method is used to correct the projection of changes in precipitation, monsoon intensity, wind and specific humidity. However, the authors don't show whether the correction of common change bias of each variable is corrected by the historical common bias from itself or not. If so, the connection between the historical bias of one variable and its changes should be demonstrated at first, which is the precondition to use the EPR method.

**Response**: As mentioned by the reviewer, it is indeed the first step to figure out whether the correction of common change bias of each variable is corrected by the historical common bias from itself or not. Due to the complexity of the EASM system, however, it is still too hard to demonstrate this. The effect of historical common bias on the common change bias for each variable remains unclear. For example, it cannot be concluded that an underestimation of the Meiyu in the historical period will lead to an underestimation of Meiyu in projection. Many previous studies have shown the underestimation of Meiyu in the historical period (e.g., Chen and Sun, 2013; Sperber et al., 2013). However, while multimodel ensemble projects an increase in precipitation around the Meiyu belt in the near future, six models that best reproduced the observed climate project decreases in precipitation over Japan and some parts of eastern China (Chen and Sun, 2013). Therefore, we attempt to make our results more convincing by showing decreased intermodel standard deviation and improved consistency in changes of many variables instead.

We have added or revised the related discussions in Section 5 Summary and Discussion as follows:

*"Due to the complexity of the EASM system, it is still too difficult to demonstrate whether the correction of common change bias of each variable is corrected by the historical common bias from itself or not. The effect of historical common bias on the common change bias for each variable remains unclear. For example, it cannot be concluded that an underestimation of the Meiyu in the historical period will lead to an underestimation of Meiyu in projection. Many previous studies have shown the underestimation of Meiyu in the historical period (e.g., Chen and Sun, 2013; Sperber et al., 2013). However, while multimodel ensemble projects an increase in precipitation around the Meiyu belt in the near future, six models that best reproduced the observed climate project decreases in precipitation over Japan and some parts of eastern China (Chen and Sun, 2013). Therefore, we attempt to make our results more convincing by showing decreased intermodel standard deviation and improved consistency in changes of many variables over the EASM region instead."*

Additional references:

Sperber, K. R., Annamalai, H., Kang, I.-S., Kitoh, A., Moise, A., Turner, A., Wang, B., and Zhou, T.: The Asian summer monsoon: an intercomparison of CMIP5 vs. CMIP3 simulations of the late 20th century, Clim. Dyn., 41, 2711-2744, 10.1007/s00382-012-1607-6, 2013.

3. The EPR method introduced by Huang and Ying (2015) depends on some parameters, for example the EOF numbers. These parameters could induce some unreal correction. Thus, it should be carefully chosen and explicitly illustrated to show it is reasonably chosen here.

4. In Section 4, the authors show the intermodel standard deviation (SD) of precipitation decreased dramatically. How about the intermodel SD of changes in specific humidity and wind?

**Response to comments #3 and #4**:

As suggested by the reviewer, we have added Table 2 in the revised manuscript, including parameters and the EASM region mean intermodel standard deviation (SD) for each variable we corrected. It is found that the mean intermodel SD is reduced largely when the original intermodel SD is relatively large, such as precipitation, 500-hPa geopotential height and zonal wind averaged from

115°E to 120°E. Mean intermodel SD changes slightly when the original intermodel SD is relatively small. We have added these analyses in Section 4 as follows:

*"Table 2 shows original and corrected EASM region mean intermodel standard deviation per degree of GMST increase for each variable. It is found that the mean intermodel standard deviation is reduced largely when the original intermodel standard deviation is relatively large, such as precipitation, 500-hPa geopotential height and zonal wind averaged from 115°E to 120°E. The mean intermodel standard deviation changes slightly when the original intermodel standard deviation is relatively small. Generally, the EPR method works well over the EASM region."*

Table 2. Parameters and the EASM region mean intermodel standard deviation (MISD) per degree of GMST increase for each variable being corrected. (N: number of models; M: number of EOF modes; Pre: Precipitation; Zg500: 500-hPa geopotential height; U850 and V850: zonal and meridional 850-hPa wind, respectively; Hus850: 850-hPa specific humidity; U115°E-120°E, V115°E-120°E and Wap115°E-120°E: meridional sections of zonal, meridional and vertical wind averaged from 115°E to 120°E, respectively; org: original data; cor: corrected data)

| Variables | Pre | Zg500 | U850 | V850 | Hus850 | U115°E-120°E | V115°E-120°E | Wap115°E-120°E |
|---|---|---|---|---|---|---|---|---|
| N | 19 | 14 | 19 | 19 | 19 | 19 | 19 | 19 |
| M | 16 | 12 | 16 | 16 | 16 | 16 | 16 | 16 |
| MISDorg1.5 | 0.47 | 3.09 | 0.39 | 0.29 | 0.21 | 0.61 | 0.40 | 0.004 |
| MISDcor1.5 | 0.13 | 0.41 | 0.43 | 0.32 | 0.25 | 0.35 | 0.52 | 0.002 |
| MISDorg2 | 0.33 | 2.96 | 0.31 | 0.20 | 0.17 | 0.47 | 0.27 | 0.002 |
| MISDcor2 | 0.08 | 0.35 | 0.29 | 0.21 | 0.18 | 0.27 | 0.35 | 0.001 |

5. This study corrects the projections of EASM change at global warming of 1.5°C and 2°C. The authors conclude the difference of EASM precipitation change between 1.5°C and 2°C is because of the moisture increase by extra 0.5°C. What is meaning to compare the EASM between global warming of 1.5°C and 2°C? The EPR method doesn't seem to help compare their differences.

**Response**: Projected climatic impacts on human lives for the 2 °C warming scenario may exceed the adaptation capacity of most vulnerable countries, such as small island nations. As such, many countries have advocated a more aggressive goal of limiting warming to less than 1.5 °C. Therefore, it is important to evaluate the differences in regional climate changes between 1.5 °C and 2 °C warming. The EPR method is believed to give us more reliable projections of the EASM change at global warming of 1.5°C and 2°C, together with their differences. In this study, a more significant change of EASM precipitation is projected using this method, suggesting that extra 0.5°C would lead to more climatic risks over East Asia in future.

6. Only 19 models were used in this study. I suggest more models should be used to increase the robustness of multi-model projection.

**Response**: As mentioned by the reviewer, results from more models will indeed increase the robustness of multimodel projection. However, we have taken another strategy toward this goal, considering a large amount of data we have to download. We first examined performances of 43 available CMIP5 CGCMs in simulating EASM precipitation for the reference period of 1986-2005. Then, the CGCMs with the skill score S (Taylor, 2001) greater than 0.75 are selected, in terms of precipitation over the EASM region (100°-150°E, 20°-50°N), which matters the most. The original MMM projections are similar to the projections by others (e.g., Chen and Sun, 2013; Jiang and Tian, 2013; Seo et al., 2013). Thus, our results are reliable when 19 relatively good models are selected. We mentioned this in Section 2.1 Data.